

# Cognitive expertise in esport experts: a three-level model meta-analysis

Haofei Miao[1,*], Hao He[1,*], Xianyun Hou[2], Jinghui Wang[1] and Lizhong Chi[1]

[1] School of Psychology, Beijing Sport University, Beijing, China
[2] School of Philosophy, Wuhan University, Wuhan, China
[*] These authors contributed equally to this work.

## ABSTRACT

**Objectives**. The cognitive expertise of experts has been an intriguing theme; there has been rapid growth in cognitive research related to esports. Given the close association between esports activities and cognition, esports holds promise in offering new perspectives for understanding cognitive expertise. This meta-analysis aims at quantitatively delineating the cognitive disparities between esports experts and amateurs.

**Methods**. The expert group comprised professional video game players and high-ranking players (top 1%), while amateurs were assigned to the control group. Research studies published between January 2000 and December 2023 were systematically searched in databases. A three-level model with cluster-robust variance estimation was used to calculate the overall effect size. The moderating variables included professional level, cognitive abilities, dependent variable type, game genre, gender and age.

**Results**. A total of 15 studies containing 142 effect sizes and 1085 participants were included in this meta-analysis. The results indicated that, compared to amateurs, video game experts demonstrated superior cognitive abilities with a small effect size (Hedges' g = 0.373, 95% CI [0.055–0.691], $p$ = .012). The differences between experts and amateur players mainly manifest in spatial cognition and attention. Sensitivity analysis, risk of bias, and publication bias results indicated the reliability of these findings.

**Conclusions**. This meta-analysis confirms that esports experts possess superior cognitive abilities compared to amateurs, particularly in aspects of spatial cognition and attention. These can provide an effective reference for future selection and training in esports.

Corresponding author
Lizhong Chi, chilizh3804@163.com

## INTRODUCTION

Esports involves competitive video game play on electronic systems like computers, tablets, or mobile phones. It features teams and individuals competing online or in local-area-network tournaments (*Pedraza-Ramirez et al., 2020*). Popular titles such as Counter-Strike: Global Offensive (CSGO), League of Legends (LOL), and Dota2 and Dota2 are usually structured with ranking systems. Esports has experienced an extraordinary surge in growth over the past two decades. It is estimated that the esports audience reached approximately 646 million at 2023 (*Newzoo, 2022*). This development has not only attracted a larger player

base but also provided abundant opportunities for investigating the expertise of high-level players.

Esports now presents a novel avenue to explore the intricates of expertise (*Campbell et al., 2018*; *Wang, Zhang & Li, 2022*; *Phillips, 2023*). Esports provide a highly conducive setting for the investigation of cognitive expertise, owing to their distinctive characteristics including a high cognitive load, ecological validity, and a relatively lower entry barrier (*Campbell et al., 2018*; *Dale et al., 2020*). Firstly, the fast-paced and competitive nature of gaming necessitates swift movements and rapid response times. Competitive video games encompass a wide array of cognitive demands such as perception, attention, memory, planning, and decision-making (*Bediou et al., 2018*). Additionally, the operational environment of video games aligns well with the cognitive paradigm rooted in computer-based tasks, thereby enhancing the ecological validity of performance explanations (*Pluss et al., 2020*). Esports players do not require extensive physical or strength training, and the accessibility of gaming equipment reduces the entry barrier to esports (*Ericsson, Krampe & Tesch-Römer, 1993*). These factors contribute to the advantages of esports in developing and researching expertise. As a valuable model for the study of cognitive expertise, esports can be likened to the role played by fruit flies and mice in the field of biomedical. Esports could provide abundant and expedited opportunities for motor and cognitive skills development (*Pedraza-Ramirez et al., 2020*; *Toth et al., 2020*).

Understanding cognitive expertise in esports necessitates a clear definition of what constitutes an expert. *Mendoza et al. (2023)* summarized three types of defining criteria were identified: (1) being a professional player, (2) being part of an organized team and (3) having experience in competitions. "Being part of an organized team" refers to being involved in various levels of organizations, ranging from university teams to professional leagues. With the advent of ranking systems in most competitive video games, a popular approach is to define experts based on their ranking instead of "their experience in competitions". Players who achieve high rankings on game servers, often measured by matchmaking rating (MMR), are considered experts. For instance, experts may be defined as the top 18%, 7%, or even the top 0.15% of players (*Qiu et al., 2018*; *Toth, Kowal & Campbell, 2019*; *Gan et al., 2020*; *Yao et al., 2020*). This approach aligns with the ranking system utilized in chess expert meta-analyses, where player' levels are distinguished based on rankings or ranking points, similar to the Elo system used for chess players (*Burgoyne et al., 2016*). *Baker, Wattie & Schorer (2015)* recommended experts might be identified as performers in the top 5%, top 10% or with scores that are two standard deviations above the mean performance for the population. With a global player base in CSGO potentially exceeds over 1 billion, the top 5% or top 10% still constitute a substantial group. Considering that so many people are labeled as experts would be unreasonable, we defined esports experts as those ranked within the top 1% and above. Additionally, if players are registered as professional players who have undergone systematic training (*Tanaka et al., 2013*; *Bediou et al., 2018*; *Ding et al., 2018*), they are undoubtedly considered experts. In summary, our definition of "experts" encompasses both professional players, or being part of an organized team and high-ranked amateur players (top 1%), covering individuals excelling in game rankings and demonstrating a high level of game skills.

Compared to amateur gamers, esports experts showcase notable advantages across various cognitive abilities. The most prominent strength of esports experts appears to reside in spatial cognitive abilities (spatial working memory, mental rotation, *etc.*) (*Tanaka et al., 2013*; *Bediou et al., 2018*; *Kang et al., 2020*). In addition, previous studies have shown that esports experts outperform amateurs in task-switching ability (*Li et al., 2020*). Concerning basic cognitive abilities, esports experts exhibit strengths in sustained attention (*Benoit et al., 2020*; *Li et al., 2020*), but results regarding inhibitory control of attention have been inconclusive (*Benoit et al., 2020*; *Valls-Serrano et al., 2022*). It is worthing that in traditional competitive sports, differences in basic cognitive abilities between experts and amateurs are also controversial. A meta-analysis by *Kalén et al. (2021)* suggests basic cognitive abilities (inhibitory control of attention, *etc.*) cannot differentiate between experts and amateur players well. This is because the effect size of basic cognitive differences between professional athletes and amateurs is small. Based on results from mainstream sports, it is anticipation and decision-making abilities that are the most distinguishing cognitive skills predicting expert performance (*Kalén et al., 2021*). However, in the realm of esports, basic visual cognitive abilities could predict game performance such as player's rankings, highlighting the potential significance of basic cognitive abilities in this esports (*Kokkinakis et al., 2017*; *Röhlcke et al., 2018*; *Large et al., 2019*; *Cretenoud et al., 2021*). This meta-analysis aims to investigate these cognitive advantages across nine domains (perception, bottom-up attention, top-down attention, spatial cognition, task-switching/multitasking, inhibition, problem-solving, verbal cognition and motor control) (*Bediou et al., 2023*), shedding light on the unique cognitive profiles of esports experts.

Meta-analyses in traditional sports have already embraced players from various sports (*Scharfen & Memmert, 2019*; *Kalén et al. (2021)*), yielding valuable insights for psychological selection and training. In line with this, our meta-analysis includes multiple types of competitive games rather than focusing on a single genre. This approach enables us to compare effect sizes across different game genres. In recent years, there has been a blurring of boundaries between different types of competitive video games, with some games combining elements from multiple genres. This is evident in games like Valorant and Overwatch, which incorporate elements of both multiplayer online battle arena (MOBA) and first-person shooter (FPS) genres. In addition, some professional gamers have excelled across different games. Similar multi-game success can also be observed in traditional sports, as seen with Steve Curry's achievements in both basketball and his natural talent for golf. Cognitive and motor expertise may be a contributing factor to this cross-game success.

The primary focus of this meta-analysis is to examine differences between experts and amateur players, rather than novice. The impact of gaming experience on cognition is evident (*Bediou et al., 2018*; *Bediou et al., 2023*; *Sauce et al., 2022*). This approach, utilizing amateur players as the control group rather than novices, allows for partial control of gaming experience discrepancies between the two groups. It enhances our understanding of cognitive expertise in esports experts by providing a nuanced exploration of cognitive differences within the context of varied game genres.

In the field of cognitive studies in esports, a cognitive ability usually includes several indicators such as speed (reaction time, RT) and accuracy. Previous meta-analyses, often favored selecting the metric with the larger effect size among multiple indicators on the same dependent variable with accuracy commonly prioritized over response time (*Bediou et al., 2018*). So previous meta-analyses based on cross-sectional controlled studies typically included only one effect size in a study, as shown in a classical forest plot (*Neguţet al., 2016*). However, this approach of selection a single metric may result in the loss of valuable information, potentially leading to inaccurate results. When different metrics are included in a study, the variance between effect sizes from the same study becomes crucial. Traditional fixed-effects and random-effects models used in meta-analysis do not adequately address this concern, as they primarily focus on between-study heterogeneity and sampling heterogeneity.

To overcome these limitations and address both within-study and between-study heterogeneity, the three-level meta-analytical model has been proposed (*Van den Noortgate et al., 2013*; *Fernández-Castilla et al., 2021*). This model allows for the inclusion of multiple effect sizes from different dependent variables and can capture heterogeneity at three levels. This approach is well-suited for cognitive studies of experts and amateur players, as these studies often capture multiple metrics for multiple cognitive tasks. In present meta analyses, specific cognitive abilities are categorized based on the cognitive classification methods of *Bediou et al. (2018)* and *Bediou et al. (2023)*.

Cognitive abilities often exhibit positive correlations to varying degrees. The utilization of cluster-robust variance estimation in the three-level model allows for the simultaneous consideration of the interdependence between effect sizes derived from the same study (*Pustejovsky & Tipton, 2022*). Therefore, the three-level model can improve the accuracy and reliability of parameter estimation and statistical inference.

This meta-analysis focusing on cognitive differences between expert and amateur gamers rather than novices. The following moderator variables will be considered: professional level, cognitive abilities, dependent variable type, game genre, and age. Employing a three-level model with cluster-robust variance estimation will facilitate the calculation of effect sizes. We hypothesize that esports experts possess superior cognitive abilities, particularly in the realm of attention and spatial cognition. The results of the study are anticipated to provide valuable insights for talent development and to understand better which are the key cognitive abilities in esports.

## METHOD

This study was registered with Open Science Framework (OSF, https://osf.io/kdfn2/), and adhered to the Preferred Reporting Items for Systematic Reviews and Meta-Analyses (PRISMA) guidelines (*Page et al., 2021*). The ethical approval was not required in this secondary data analysis.

A total of 2,455 articles were initially identified through the searches conducted. After a thorough investigation and exclusion of irrelevant articles, 15 studies were included in the meta-analysis. The most common esports genres were Strategy games (studies

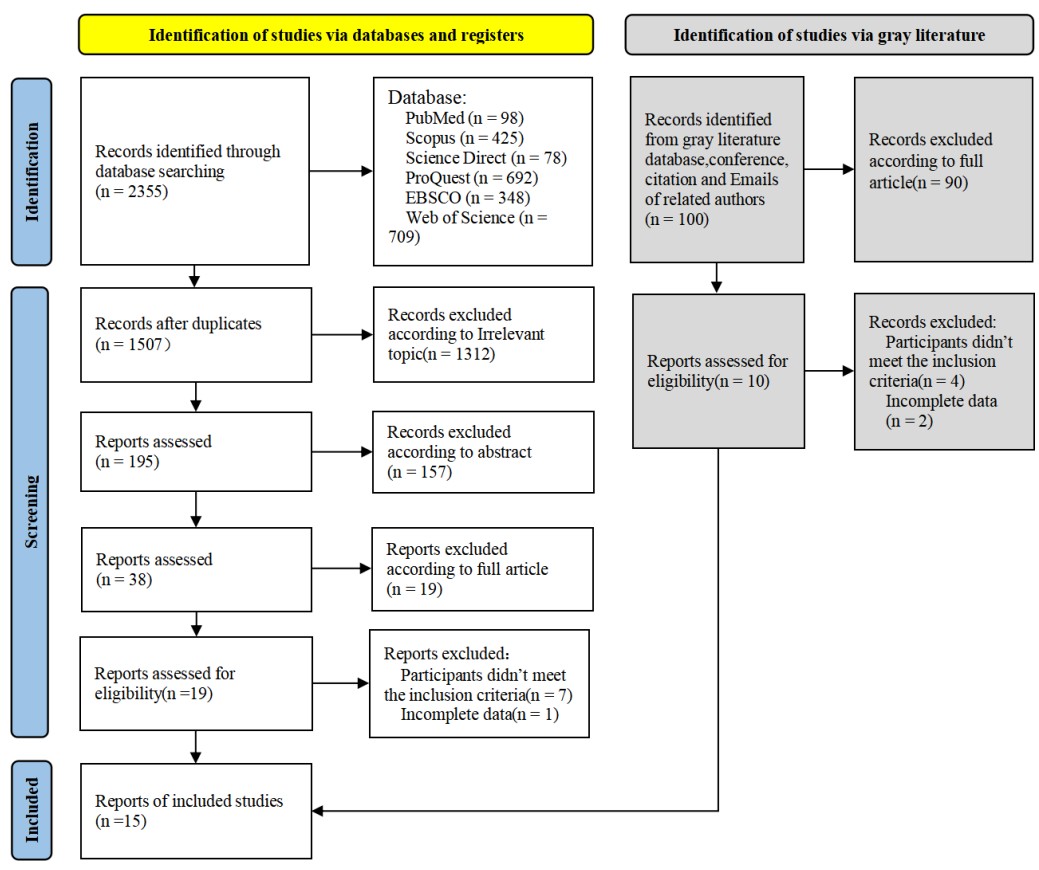

**Figure 1    PRISMA flow diagram.**

$m = 6$) and Shooting games (studies $m = 2$), while some studies combined several game genres were included Mix games genres (studies $m = 5$). The remaining research was included in the "Other games" category (studies $m = 2$). See Table S3 for specific types of games. In 12 studies, the expert group consisted of professional players, while in the other studies, the expert group comprised highly ranked non-professional players. Ultimately, the meta-analysis analyzed a total of 142 effect sizes from 15 articles, with a combined sample size of 1,085 participants. A PRISMA flow diagram of the selection process is presented in Fig. 1. Characteristics of studies included in this meta-analysis was shown in Table 1.

## Search strategy

The literature search encompassed the timeframe spanning from January 2000 to December 2023. We conducted inquiries across several databases, namely PubMed, Web of Science, Scopus, Science Direct, ProQuest, EBSCO (Academic Search Complete, SocINDEX with Full Text, APA PsycInfo, APA PsycArticles, Psychology and Behavioral Sciences Collection, OpenDissertations, ERIC, APA PsycTests, APA PsycTherapy), utilizing terms merged within the ensuing Boolean expression (cognition OR cognitive OR perception OR perceptual OR attention OR attentional OR working memory OR executive function OR executive control OR inhibition OR task switching OR multitasking OR Multiple-Object

**Table 1  Characteristics of studies included in the meta-analysis.**

| Num | Study | Games | Purpose of study | Participants (n) | Main criteria for experts | Main criteria for controls | Cognitive/motor task | Gender ratio of experts vs. controls (male/female) | Mean age of experts vs. controls | First auther country |
|---|---|---|---|---|---|---|---|---|---|---|
| 1 | *Benoit et al. (2020)* | OW, CS, COD,Rainbow 6, PUBG, Battlefield 4 | Expertise differences | Professional players(14) vs.Amateurs(16) | From league clubs and with top 500 in game rankings or Grandmaster ranking tie. | College students with unlimited game rankings. | Attention, WAIS-IV Coding, WMS-III Spatial Span, WAIS-IV Digit Span, D-KEFS Towers, D-KEFS Color-Word, Grooved Pegboard. | 14/0 vs.12/4 | 23.66 vs. 25.31 | Canada |
| 2 | *Bickmann et al. (2021)* | MOBA, FPS | Expertise differences | Professional players(18) vs.Amateurs(21) | Players who make a living from video gaming. | College students who are not making money off of gaming. | SVRT, SART, Choice Reaction time. | 17/1 vs. 19/2 | 22.83 vs. 23.86 | German |
| 3 | *Cretenoud et al. (2021)* | CSGO | Effects of esports | High-ranking amateurs(15) vs. Low-ranking amateurs(79) | Top 1% with global elite rank tier. | Ranked below 1%. | 1-back, SRT, Freiburg visual acuity, VS. | All male | Total 21.9 | Switzerland |
| 4 | *Ding et al. (2018)* | LOL | Expertise differences | Professional players(10) vs. Semi-professional trainees(10) vs. Amateurs(20) | Professional and trainee players both from esports gaming clubs. | College students with 300 or more games played. | SRT, flanker task, VS, MOT, motor control, | All male | 21 vs.18 vs. 20 | China |
| 5 | *R Core Team (2022)* | LOL; CSGO | Expertise differences | Professional players(10) vs. Amateurs players(10) vs. Novice(5) | From two professional e-sport teams. | Never competed in professional game. | Change detection test, Go/No-go, Iowa Gambling Task, MR, TOL, Rotation, Situation Awareness, Time Wall, VS, Dexterity, APM, Pursuit Rotor. | All male | 21.5 vs.24.9 vs. 28.4 | Sweden |
| 6 | *Gao (2019)* | LOL | Expertise differences | Professional players(18) vs. Amateurs(20) vs. Novice(17) | From the Collegiate League Championship Team. | Amateur: college students who didn't participate in tournaments; Novice: college students who never or very little video game experience. | SRT,movement anticipation,VS. | 15/3 vs. 16/4 vs. 11/6 | 21.25 vs. 20.56 | China |
| 7 | *Kang et al. (2020)* | LOL, OW,BG,SC | Expertise differences | Professional players(55) vs. Amateurs(60) | From professional e-sport teams. | Healthy control who gameplay time less than 30 h/week. | TOL, MR. | All male | 21.3 vs. 21.3 | Korea |
| 8 | *Kim, Kim & Wu (2023)* | SC | Expertise differences | Professional players(8) vs. Amateurs(8) | From professional e-sport teams. | College amateurs who have never played a strategy game. | Anticipation timing, Peripheral Perception, Eye movement and coordination. | All male | 23.8 vs. 24.7 | Korea |
| 9 | *Li et al. (2020)* | LOL | Expertise differences | High-ranking amateurs(35) vs. Low-ranking amateurs(35) | Ranked in the top 0.15% (Master and above). | Ranked after 0.15% (Iron-Diamond). | Continuous performance test, Stroop-switching task. | 33/2 vs. 31/4 | 22.8 vs. 23.1 | China |
| 10 | *Phillips (2023)* | Fighting; Rhythm Games | Expertise differences | Professional players(49) vs. Amateurs(37) | From professional e-sport teams. | Ranked after 1%. | UCMRT, Serial Reaction Task, CBTT, SVRT, SART, Paced motor timing , Sustain attention. | All male | 23.88 vs. 22.94 | USA |
**Table 1** (*continued*)

| Num | Study | Games | Purpose of study | Participants (n) | Main criteria for experts | Main criteria for controls | Cognitive/motor task | Gender ratio of experts vs. controls (male/female) | Mean age of experts vs. controls | First auther country |
|---|---|---|---|---|---|---|---|---|---|---|
| 11 | *Pluss et al. (2020)* | LOL,HOS,OW,PUBG | Expertise differences | Professional(25) vs. Recreational(25) vs. Novice(25) | From professional e-sport teams. | Amateurs see game as leisure activity; novice with no experience in esports. | SRT, Selective reaction time, Manual dexterity, switch task. | 25/0 vs. 21/4 vs. 18/7 | 22.05 vs. 25.80 vs. 24.69 | Australia |
| 12 | *Röhlcke et al. (2018)* | DOTA2 | Effects of esports | High-ranking amateurs(17) vs. Low-ranking amateurs(287) | Ranked in the top 1%. | Ranked after 1%. | WM Operation Span,WM Spatial Span, WM Digit Span. | All male | 23.35 vs. 22.41 | Sweden |
| 13 | *Tanaka et al. (2013)* | Guilty Gear | Expertise differences | Professional players(17) vs. Amateurs(33) | Participate in international or national competitions. | Had negligible or no video game experience. | Visual WM Task. | All male | 24.1 vs. 22.4 | Japan |
| 14 | *Valls-Serrano et al. (2022)* | LOL | Expertise differences | Professional players(20) vs. Amateurs(30) vs. Novice(20) | From the UCAM esports club primary and secondary team and in top 1% ranking. | Amateurs: Ranked after 1% and MOBA is their main game. Novice was playing video games less than 1 h per week. | Antisaccade Task, Number-Letter Task, Stop Signal Task, CBTT. | All male | 21.8 vs. 22.0 vs. 22.9 | Spain |
| 15 | *Wang, Zhang & Li (2022)* | LOL, PUBG, PUBG: Mobile, Artifact, Popkart. | Expertise differences | Professional players(33) vs. Amateurs(48) | Participate in a primary and secondary gaming league. | Casual players who are not participating in open tournaments. | 2D-MOT, 3D-MOT. | Both groups contain males and females | 22.7 vs. Un-record | China |

**Notes.**

Game, Heroes of the storm: BG, Battleground; SC, StarCraft; LOL, League of Legends; OW, Overwatch; CS, Counter Strike; CSGO, Counter Strike: Global Offensive; COD, Call of Duty; Player Unknown's Battlegrounds. cognitive task.

Cogation: APM, Action per minute; CBTT, Corsi block tapping test; MOT, Multi-object tracking; MR, Mental Rotation; UCMRT, University of California Matrix Reasoning Task; VS, Visual search; WM, working memory; SART, Simple Acoustic Reaction Time; SRT, Simple reaction time; SVRT, Simple Visual Reaction Time.

Tracking OR processing speed OR reasoning OR planning OR problem solving OR decision making) AND (experts OR professional players OR top players OR elite players OR gamers) AND (video game OR computer game OR esports). In the event that a Boolean search was not permissible, we endeavored to execute alternative search combinations. Other details of our search strategy referenced the meta-analysis about the cognitive abilities in action video game players or athletes (*Bediou et al., 2018*; *Bediou et al., 2023*; *Kalén et al., 2021*).

The search method for gray literature was based on the approach used by *Bediou et al. (2018)*: (a) Queried specialized databases for gray literature such as PsycEXTRA, ScholarOne, opengrey, and base-search. (b) Searches were conducted within the abstracts of annual conferences. (c) Additional searches were performed using Google Scholar and the Dissertations or thesis Abstracts database. (d) Based on the presently available literature, we had identified some experts in esports' expertise and attempted to establish communication with individuals, compiling a contact list for communication purposes.

Eventually, we directly contacted 20 authors in this field and inquired about any incomplete or unpublished data related to video games cognitive expertise. Subsequently, we received seven responses out of the 20 requests and obtained the requested data in four cases which met our inclusion criteria.

## Selection criteria

The inclusion criteria were: (1) studies comparing cross-sectional cognitive differences between esports experts and amateur players. (2) Experts were either registered professional players, part of an organized team, or high-ranked non-professional players (ranking ≤top 1%). (3) Amateurs had never played in tournaments and possessed lower rankings than experts, and novice was excluded. (4) The inclusion criteria of game genre was parallel with *Pedraza-Ramirez et al. (2020)*, including FPS, TPS, RTS, Fight games, Sports games, MOBA, Mobile, or other game genre (*e.g.*, card games). (5) At least one paradigm was employed to assess cognitive abilities, also including simple reaction time or motor control (*Bediou et al., 2023*). (6) Age ranged from 18 to 35, Gender was not limited.

## Data coding

The data coding section was primarily designed to collect and align the metrics of cognitive tests from different studies in order to calculate effect sizes. The study focused on cognitive ability as the dependent variable, various metrics, such as reaction time (RT) and accuracy (ACC). The RT and ACC of Cognitive tasks' means, standard deviations, and sample sizes of each group were recorded.

This meta-analysis aims at comparing the cognitive differences between experts and amateur players. In studies with three or more subject groups, such as the research conducted by *Valls-Serrano et al. (2022)*, the novice group was excluded, leaving only the expert and amateur groups. Ultimately, all studies included in present analysis featured a single expert group and a corresponding group of amateur players.

In cases where the data was unavailable, pertinent statistics were obtained from the article or extracted using GetData Graph Digitizer 2.26. The extraction and coding of the data were independently carried out by the two authors, and any discrepancies were resolved through discussion until consistent data were obtained.

## Moderators

In meta-analysis, studies typically include many moderators, which can be categorical variables (such as classifications of cognitive abilities) or continuous variables (such as age). Setting categorical variables as moderators allows for observing differences in effect sizes between different subgroups, thereby clarifying the main sources of the overall effect size. For continuous moderator variables, moderator analysis is conducted through meta-regression to test whether the coefficient of the variable significantly influences the overall effect size. The following factors were identified as moderators: professional level, cognitive abilities, dependent variable type, game genre, gender, and age.

### Professional level

The experts in this study include professional players and top 1% ranked amateur players. Therefore, the level of expertise was used as moderating variables, which were to compare the effect sizes of professional and highly ranked amateur players.

### Cognitive abilities

The classification of cognitive abilities followed *Bediou et al. (2023)* into nine classifications: perception, bottom-up attention, top-down attention, spatial cognition, task-switching/multitasking, inhibition, problem-solving, verbal cognition and motor control (see Table S1 for details).

### Dependent variable type

The types of dependent variables encompassed speed (RT) and accuracy measurements.

### Game genre

Game genres were categorized as Shooting games, Strategy games, Mix games or Other game genres. MOBA and RTS are categorized as Strategy games. FPS and TPS are categorized as shooting games. For studies that contained both shooters and also strategy games, we categorized them in the Mixed Games. Studies that could not be categorized in the above categories were placed in the "Other games" category.

### Gender

The gender variable represented the proportion of males included in the study. We incorporating gender as a moderator and conducting the meta-regression.

### Age

Age was a continuous variable and incorporated as a moderator to conducting the meta-regression.

## Data analysis

As the studies varied significantly in design and multiple effect sizes were extracted, we used three-level meta-analytical models with cluster-robust variance estimation to calculate the total effect size (*Fernández-Castilla et al., 2021*; *Pustejovsky & Tipton, 2022*). In the three-level models, the basic sampling variance (level 0) is considered, along with estimates of within-study variance ($\tau^2_{\text{within-study}}$) and between-study variance ($\tau^2_{\text{between-study}}$) representing the heterogeneity of multiple effect sizes within a study and

across different studies, respectively. In the context of comparing expert and amateur players, $\tau^2_{\text{within-study}}$ represents the variance of multiple measures of the dependent variable of the same study, such as variance from accuracy and reaction time. By considering both $\tau^2_{\text{within-study}}$ and $\tau^2_{\text{between-study}}$, the three-level model provides comprehensive information for understanding overall effect sizes and identifying sources of heterogeneity. All models were fitted using the R package metaphor, and the robust variance was estimated using the clubSandwich package (*Viechtbauer, 2010*; *Pustejovsky & Tipton, 2022*).

Considering the relatively small sample size of the included studies, Hedges' g was employed as the effect size measure. Interpreting the effect sizes, a small effect size was defined as $\leq 0.20$, a range of 0.21 to 0.50 denoted small to medium, a range of 0.51 to 0.80 indicated a medium effect size, and a large effect size was considered as $> 0.80$ (*Lachenbruch, 1989*).

## Sensitivity analysis

Sensitivity analysis was mainly used to assess the stability of the results. In order to rigorously test the stability of the results of this study, this study implemented two distinct ways: the leave-one-out method and the quartile-based analysis. The leave-one-out method was applied by omitting one specific study in each iteration, subsequently reassessing the overall effect size and its significance to gauge any potential shifts in the meta-analytic conclusions (*Borenstein, 2009*).

In the quartile-based approach, the lower and upper quartiles of effect sizes, referred to as FL and FU, were computed. Points outside the range [FL - 1.5 * (FU - FL), FU + 1.5 * (FU - FL)] were categorized as outliers. The comparison of effect size metrics, with and without these outliers, aimed to confirm the results' resilience. An insignificant variance would suggest the robustness of the findings.

## Risk of bias

Risk of bias was mainly used to assess the quality of studies included in the meta-analysis. Utilizing the Cochrane tool for bias risk evaluation, each study was individually assessed (*Higgins et al., 2019*). This task was jointly performed by two autonomous reviewers, with any disagreements resolved through discussion until a unified decision was achieved. The assessment encompassed six facets: participant selection, exposure measurement, outcome assessment blinding, completeness of outcome data, reporting selectivity of outcomes, and confounding factors. A mixed-effects meta-regression was employed to examine if these six quality indicators of the studies influenced the overall effect size, thus exploring the possibility of quality-based effect moderation.

## Publication bias

Publication bias was primarily assessed to determine whether the meta-analysis may have included an excessive number of studies reporting positive results. For publication bias investigation, two methodologies were adopted: visual funnel plot analysis and p-curve assessment (*Simonsohn, Nelson & Simmons, 2014*). The former is a commonly used method to detect publication bias, and the latter is used as a complementary method. Asymmetry in funnel plots was an indicator of publication bias. This visual examination was further

augmented by Egger's linear regression test (*Egger et al., 1997*), which statistically evaluated the asymmetry in funnel plots.

The p-curve analysis, meanwhile, was focused on detecting selective reporting or "p-hacking" by charting the distribution of significant *p*-values and analyzing the curve's skewness (*Simonsohn, Nelson & Simmons, 2014*). A p-curve that demonstrates a rightward skew is indicative of substantial evidential worth in the meta-analysis outcomes. Conversely, a leftward skew or a curve exhibiting symmetry implies a lack of such evidential substance. The determination of this evidential value was systematically conducted through the application of binomial and continuous methodologies for significance testing.

All statistical analyses were conducted using R software, Version 4.2.3 (*R Core Team, 2022*), and a significance threshold was maintained at 0.05, with 95% confidence intervals duly reported.

# RESULTS

## Overall effect size

The overall effect size of the meta-analysis was found to be significant (*Hedges' g* = 0.373, *SE* = 0.148, 95% CI [0.055–0.691], *p* = .012, *k* = 142, *m* = 15, *N* = 1085). This indicates that video game experts exhibit superior cognitive abilities compared to amateurs. Individual study effect size estimates were shown in Fig. 2.

Regarding heterogeneity, there was between-study heterogeneity ($\tau^2_{\text{between-study}}$ = 0.285, 95% CI [0.093–0.822]) and within-study heterogeneity ($\tau^2_{\text{within-study}}$ = 0.163, 95% CI [0.106–0.245]). These low heterogeneity (both between-study and within-study) was closed to previous study (*Kalén et al., 2021*). The Q test for heterogeneity was significant (*Q* = 542.426, *df* = 141, *p* < .001). These findings suggest the presence of heterogeneity in this meta-analysis, and the main source of heterogeneity attributed to differences between studies.

## Moderator analysis

The results of Moderator analysis present at Table 2.

### Professional level

Incorporating professional level as a moderator and conducting the Wald test yielded a non-significant moderation effect on the overall effect size (*F* (1, 3.1) = 2.67, *p* = .198), indicating that professional level does not significantly moderate the observed effects. However, the effect size for cognitive differences between professional players and amateur players was significant (*Hedges' g* = 0.462, *m* = 12, *k* = 102, 95% CI [0.136–0.789], *p* = .006), while the difference between high-ranked players and amateur players was not significant (*Hedges' g* = 0.027, *m* = 3, *k* = 36, 95% CI [−0.617–0.670], *p* = .935). This suggested that the overall effect size of this meta-analysis was mainly contributed by the former.

### Cognitive abilities

When incorporating cognitive abilities as a moderator, a significant moderation effect on the overall effect size was observed ($\chi^2_{(8)}$ = 5.49, *p* < .001). Among the nine cognitive

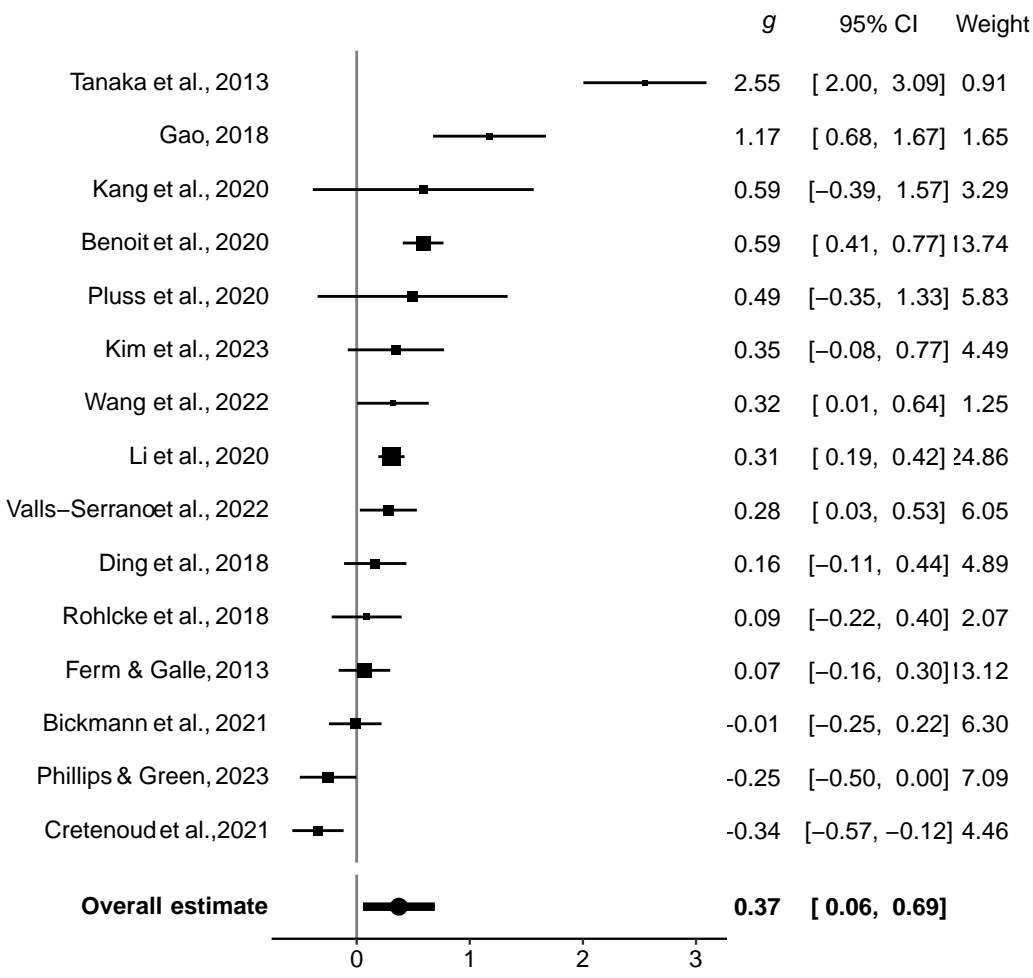

**Figure 2** **Forest plot.** The size of the square at the midpoint of each study's line represents its weight in the meta-analysis. The longer the line, the larger the confidence interval (CI) associated with the study (*Tanaka et al., 2013*; *Gao, 2019*; *Kang et al., 2020*; *Benoit et al., 2020*; *Pluss et al., 2020*; *Kim, Kim & Wu, 2023*; *Wang, Zhang & Li, 2022*; *Li et al., 2020*; *Valls-Serrano et al., 2022*; *Ding et al., 2018*; *Röhlcke et al., 2018*; *R Core Team, 2022*; *Bickmann et al., 2021*; *Phillips & Green, 2023*; *Cretenoud et al., 2021*).

classifications, spatial cognition exhibited the largest effect size (*Hedges' g* = 0.822, *m* = 8, *k* = 14, 95% CI [0.443–1.201], *p* < .001), while bottom-up attention also showed a significant effect size (*Hedges' g* = 0.416, *m* = 9, *k* = 35, 95% CI [0.101–0.732], *p* = .010). Effect sizes for other cognitive abilities were not found to be significant. What's more, including cognitive abilities in this moderator analysis reduced the between-study heterogeneity ($\tau^2_{between\text{-}study}$ = 0.232, 95% CI [0.075–0.681]). This suggests that measuring different cognitive abilities across studies is one source of heterogeneity.

### *Dependent variable type*

As for the dependent variable type measured, experts outscored amateurs more on dependent variable with accuracy compared to speed, $g_{accuracy}\text{-}g_{speed}$ = 0.345, $F_{(1, 9.3)}$ = 6.53, *p* = 0.030. Specifically, the effect size of accuracy was significant (*Hedges' g* = 0.560,

**Table 2 The differences between the levels of the moderator and the corresponding effect sizes.**

| Moderator-Level | Hedges' g | 95% CI | m | k | p |
|---|---|---|---|---|---|
| **Professional level** | $F(1, 3.1) = 2.67$ | | | | .198 |
| Professionals vs. Amateurs | 0.462 | 0.136, 0.789 | 12 | 106 | .006** |
| High-ranked vs. Amateurs | 0.027 | −0.617, 0.670 | 3 | 36 | .935 |
| **Cognitive abilities** | $\chi^2(8) = 5.49$ | | | | <.001*** |
| Perception | −0.138 | −0.609, 0333 | 5 | 10 | .566 |
| Bottom-up attention | 0.416 | 0.101, 0.732 | 9 | 35 | .010** |
| Top-down attention | 0.447 | −0.088, 0.982 | 4 | 6 | .101 |
| Spatial cognition | 0.822 | 0.443, 1.201 | 8 | 14 | <.001*** |
| Task-switching/multitasking | 0.266 | −0.232, 0.764 | 2 | 25 | .295 |
| Inhibition | 0.171 | −0.283, 0.625 | 4 | 9 | .461 |
| Problem solving | 0.194 | −0.210, 0.598 | 4 | 16 | .347 |
| Verbal cognition | 0.425 | −0.122, 0.972 | 2 | 6 | .128 |
| Motor control | 0.255 | −0.114, 0.624 | 6 | 21 | .176 |
| **Dependent variable type** | $F(1, 9.34) = 6.53$ | | | | .030 |
| Accuracy | 0.560 | 0.265, 0.855 | 14 | 62 | <.001*** |
| Speed | 0.215 | −0.072, 0.502 | 12 | 80 | .143 |
| **Game genre** | $F(3, 1.55) = 0.098$ | | | | .953 |
| Shooting games | 0.141 | −5.852, 6.134 | 2 | 27 | .815 |
| Strategy games | 0.382 | 0.004, 0.759 | 6 | 58 | .152 |
| Mix games | 0.274 | −0.046, 0.594 | 5 | 47 | .152 |
| Other games | 0.950 | −16.684, 18.583 | 2 | 10 | .815 |
| **Gender (male ratio)** | $\beta = 1.981$ | −3.361, 7.323 | 15 | 142 | .467 |
| **Age** | $\beta = 0.163$ | 0.021, 0.305 | 15 | 142 | .024* |

Notes.
Each row corresponds to a specific level of the moderating variable and displays the corresponding effect size and significance.
k, number of effect sizes; m, number of clusters ($F$ tests) or individual studies ($t$ tests) for each moderator and each level.
*$p < .05$.
**$p < .01$.
***$p < .001$.

$m = 14$, $k = 62$, 95% CI [0.265–0.855], $p < .001$), while the speed was not significant (*Hedges' g* = 0.215, $m = 12$, $k = 80$, 95% CI [−0.072–0.502], $p = .143$).

## Game genre

When considering game genre as a moderator and performing the Wald test, there was no significant moderation effect on the overall effect size ($F(3, 1.55) = 0.098$, $p = .953$). This suggests that game genre does not play a significant role in this context. The effect sizes of the three game genres were also not found to be significant.

## Gender

As the moderator of gender ratio was a continuous variable, the moderating effect was interpreted using meta-regression and post-hoc simple slope analysis. The results showed a non-significant moderation effect ($\beta = 1.981$, $SE = 2.725$, $p = .467$).

*Age*

When incorporating age as a moderator and conducting the meta-regression, a significant moderation effect on the overall effect size was found ($\beta = 0.163$, $SE = 0.072$, $p = .024$), indicating that age exerted a significant positive effect in this context.

### Sensitivity analysis

The leave-one-out analyses employed in our research affirmed the consistency of our findings. By sequentially omitting each study and recalculating pooled estimates, these consistently fell within the 95% confidence interval of the collective estimate. Notably, the direction and statistical significance of these recalculated results did not deviate (all *ps* > .05), underscoring the dependability and steadiness of our study's conclusions.

In our quartile outlier analysis, the comparison of effect sizes, with and without outliers, resulted in an insignificant difference ($z = 0.873$, $p = .383$), bolstering the credibility and stability of the outcomes we obtained.

### Risk of bias

In order to assess potential study risk of bias, moderating analyses were conducted using all six indicators of study quality included as moderators. None of the moderating effects were found to be significant. Specifically, the moderating effects of confounding variables ($F (2,2.32) = 1.33$, $p = .412$), blinding of outcome assessments ($F (1,5.26) = 0.844$, $p = .398$), incomplete outcome data ($F (1,10.1) = 1.15$, $p = .309$) were not significant. No sensitivity analysis was run for the selection of participants, measurement of exposure and incomplete outcome data, as all studies had the same classification. These findings suggest that the results of this meta-analysis were not influenced by the quality of the individual eligible studies. The quality of the included studies is acceptable.

### Publication bias

To assess the presence of reporting bias, the symmetry observed in the funnel plot (illustrated in Fig. 3) was examined. The symmetry was corroborated by the results from Egger's linear regression test (precision = $0.428 \pm 1.072$, $t (3.61) = 0.399$, $p = .712$). Additionally, the application of the trim-and-fill method indicated no requirement for imputed studies to rectify publication bias. The results of the p-curve analysis, depicted in Fig. 4, showed a rightward skew. The statistical power derived from this analysis was calculated to be 87% (90% CI: 78%–93%). Evaluations for rightward skewness, employing both binomial and continuous statistical approaches, validated the evidential substance in our meta-analysis findings. The binomial test showed a significant outcome ($p = .008$), and similar significance was observed in continuous analyses, both for the full p-curve ($z = -10.36$, $p < .0001$) and the half p-curve ($z = -11.73$, $p < .0001$). Collectively, these analyses consistently indicated an absence of publication bias in present study.

## DISCUSSION

### Main finding

This meta-analysis evaluates the difference of cognitive abilities in esports experts compared to amateurs. The results of three-level meta-analytical models with cluster-robust variance
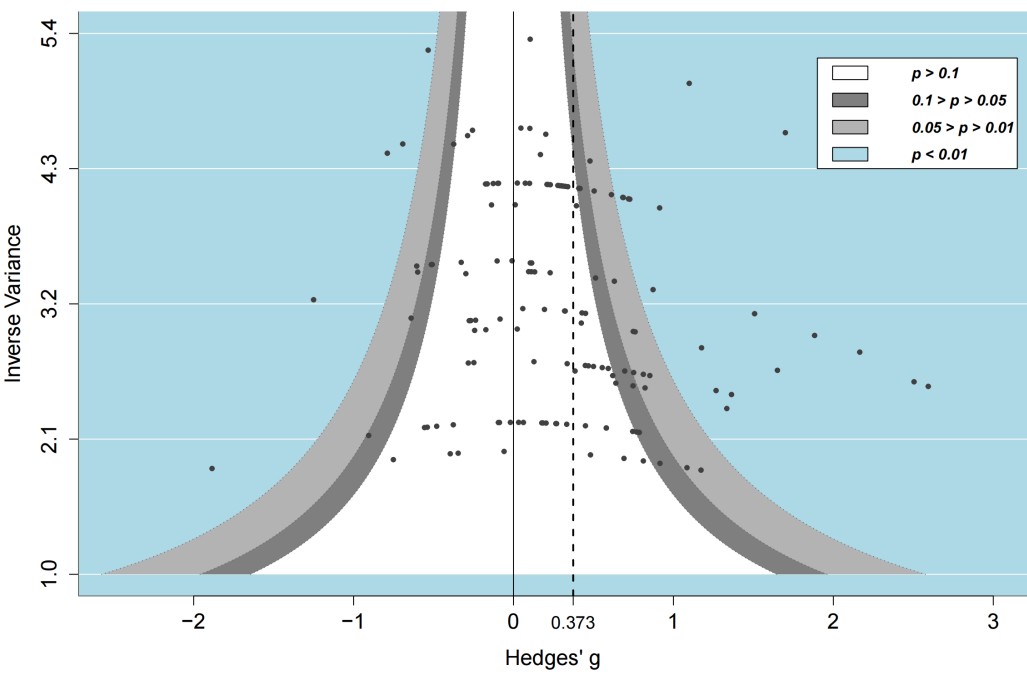

**Figure 3    Funnel plots for all effect sizes.**

estimation indicate that the overall effect size is significant and its magnitude is small to medium (*Hedges' g* = 0.373), illustrating that experts did exhibit superior cognitive abilities than amateurs. Specifically, the effect size of cognitive differences between professional players and amateurs (*Hedges' g* = 0.462) was larger than that between high-ranked players and amateurs (*Hedges' g* = 0.027). Among the nine cognitive classifications, the cognitive expertise of esports experts is mainly reflected in spatial cognition and bottom-up attention. The minor but primary origin of heterogeneity arises from between-study discrepancies, although the overall degree of heterogeneity is low. Through sensitivity analyses, assessment of bias risk, and publication bias testing, the current findings have demonstrated robustness, affirming the reliability of the results obtained in this meta-analysis.

## Different effects of moderators

The effect size for cognitive disparities between professional players and amateurs is moderately small to medium (*Hedges' g* = 0.462), whereas the distinction between high-ranked players and amateur players lacks significance (*Hedges' g* = 0.027). This suggests that the overall effect size in this meta-analysis predominantly stems from the former group. However, the moderating effect of professional levels is not significant. This could be attributed to the inclusion of 12 articles on professional players and only three articles on high-ranked players in this meta-analysis. With the limited number of studies on the latter, conclusive inferences cannot be drawn. Presently, a confident assertion can be made that professional players demonstrate noteworthy cognitive advantages over amateur players.

The role of various cognitive abilities differs notably within the realm of esports. We classified cognitive abilities into nine classifications: perception, bottom-up attention,

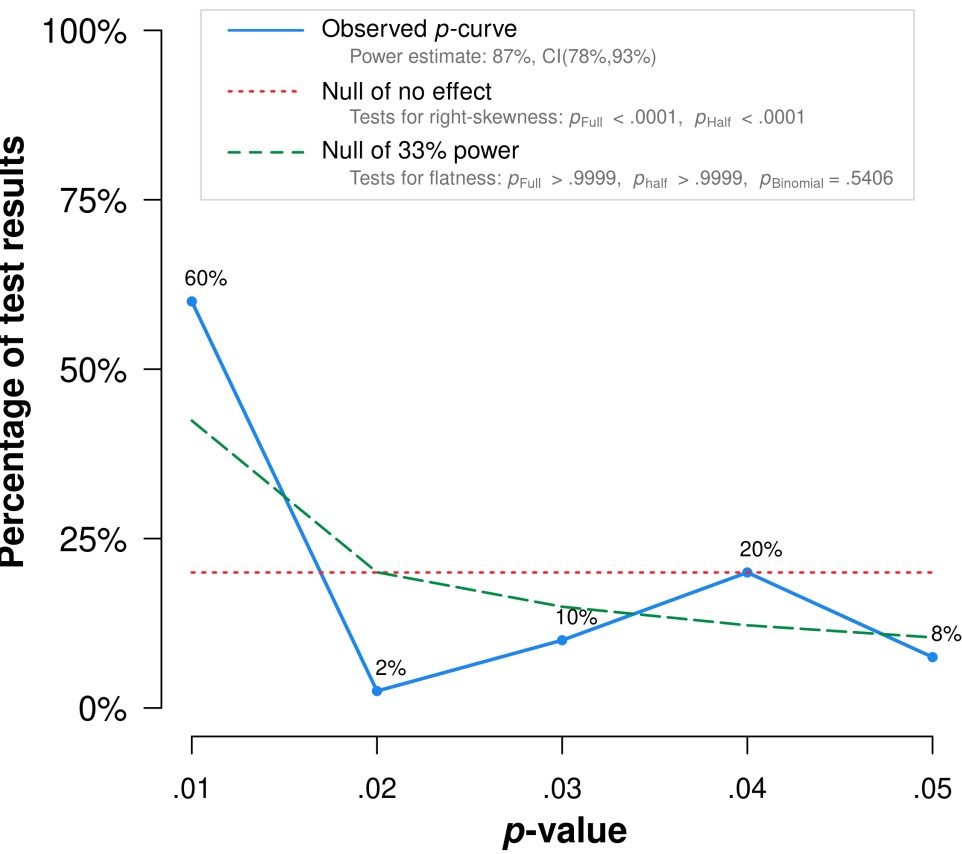

Note: The observed *p*-curve includes 40 statistically significant (*p* < .05) results, of which 28 are *p* < .025. There were 102 additional results entered but excluded from *p*-curve because they were *p* > .05.

**Figure 4** P-curve plot showing the distribution of significant results (*p* < .05) in the present pool of eligible reports.

top-down attention, spatial cognition, task-switching/multitasking, inhibition, problem-solving, verbal cognition and motor control. Spatial cognition exhibits the largest effect size (*Hedges' g* = 0.822). Among all these nine cognitive abilities, spatial cognition and bottom-up attention present significant effects, while none of the other abilities are statistically significant. Such a result indicates that spatial cognition and attention contribute significantly to the overall effect size of this study. The attention and spatial cognition advantages of experts have a neurobiological basis. Specifically, experts exhibited enhanced intra- and internetwork functional integrations between the salience network (SN) and central executive network (CEN) (*Gong et al., 2016*). Within spatial cognition, the most frequently used test is visual working memory. Actually, remembering and updating the spatial positions of multiple opponents and skill cooldown times within the game scenario are crucial. Hence, to some extent, spatial working memory might even be considered a gold standard for talent assessment in esports. Selective reaction time falls under the classification of bottom-up attention. In numerous gaming scenarios, players are required to be selective in their actions based on their opponents' moves. This often

involves selective reaction time. Notably, in terms of motor control, experts do not exhibit a significant advantage. This seems contradictory to our general perception of esports players. In fact, referencing the famous Chunking tasks of *Chase & Simon (1973)*, *Baba (1993)* ingeniously confirmed through a series of experiments that experts' superior motor control heavily depends on context and the gaming control tool. Once a new control tool is introduced, the experts' motor control advantage becomes less apparent. We speculate that experts' superiority in motor control is specialized and might not be pronounced in general cognitive tasks.

How are players' cognitive expertise formed? It's important to note that esports players' cognitive expertise in spatial cognition is inevitably influenced by long-term gaming experience. According to *Bediou et al. (2018)* and *Bediou et al. (2023)*, gaming training has a substantial impact on spatial cognition (*Hedges' g* $= 0.26 \sim 0.45$). However, it is evident that current gaming training researches is insufficient to fully explain the large effect size observed between experts and amateurs. This suggests that these players may have already possessed certain cognitive advantages before attaining expert status. Interestingly, superior early cognitive abilities might lead to better performance in early gaming stages. For example, in the study by *Jakubowska et al. (2021)*, which focused on CSGO game training over a period of 28 days, novices' amplitude of the P300 component in the Attentional Blink (AB) task appeared to be predictive of the level of achievement in later games. A higher P300 wave amplitude signifies better attentional abilities (*Vogel, Luck & Shapiro, 1998*). This electroencephalographic evidence supports the notion that experts initially possess distinct cognitive neural characteristics.

Age emerges as an important moderating variable for cognitive abilities. In this study, the findings reveal a weak positive moderating effect ($\beta = 0.161$) of age on the cognitive differences between experts and amateurs. This implies that as players' age increases, the differences between experts and amateurs become more pronounced. Notably, participants in this study were predominantly aged between 20 and 25, a stage where cognitive plasticity remains relatively high (*Bediou et al., 2023*). The increase in age implies more gaming training for professional players, thereby intensifying the differences between the two groups. Additionally, we did not find any significant moderating effects of gender. This could be attributed to the fact that current studies on cognitive expertise in esports predominantly involve male participants or have a majority of males with few females. Furthermore, notable differences among various game genres were not observed. These findings are consistent with previous research indicating no distinctions in attention and multi-target tracking abilities between FPS and MOBA amateurs (*Bickmann et al., 2021*). Perceived differences between players in different esports disciplines are still noteworthy (*Phillips, 2023*; *Phillips & Green, 2023*). Considering the early stage of esports research, we recommend researchers to thoroughly report and match the basic information of age, education and training experience between expert and control groups as comprehensively as possible.

Dependent variable type does exert a significant moderating effect. The effect size for accuracy is significant (*Hedges' g* $= 0.560$), while the reaction time is not significant. Previous meta-analyses often select either speed or accuracy metrics of cognitive tests based

on their effect size (*Bediou et al., 2018*). This approach may result in effect size inflation by focusing solely on the most prominent dependent variables and disregarding others. Consequently, we recommend that future studies report diverse metrics, such as the rate of correct rejections and false alarm rate, or consider metrics that account for the trade-off between speed and accuracy. Employing a three-level model and incorporating multiple effect sizes for the same cognitive test can enhance the quality of meta-analytical outcomes (*Fernández-Castilla et al., 2021*).

## LIMITATIONS

This meta-analysis encounters limitations due to relatively small sample sizes across most studies. Furthermore, there may be a dose effect on the impact of game training experience on cognitive expertise, but existing studies are inconsistent in reporting indicators of years and training duration for various types of games (*Smith et al., 2020*). We recommend that future research reports include the gaming experience, weekly gaming hours, and educational background of both professional and amateur players. We recommend that future studies explicitly report whether a player's ranking were achieved in the current season or in the past. Also, future studies should consider factors such as individual differences, cultural backgrounds, and social environments (*Monteiro Pereira et al., 2022*), which may play a significant role in the mental health and cognitive abilities of esports players. Finally, our meta-analysis did not include studies specifically on anticipation and decision-making abilities. Current esports studies predominantly emphasize domain-general cognitive abilities and often neglect pivotal domain-specific cognitive aspects such as anticipation or decision-making abilities (*Lindstedt & Gray, 2019*; *Delmas, Caroux & Lemercier, 2022*).

## CONCLUSION

This meta-analysis investigates cognitive abilities between esports experts and amateurs. Our findings reveal a small to medium effect size in their cognitive disparities, emphasizing the significant role of both spatial and attentional cognitive abilities in talent development. Future research should delve deeper into domain-specific skills, particularly anticipation and decision-making.

## PRACTICAL IMPLICATIONS

We have summarized four suggestions: (a) Utilizing cognitive assessments for esports selection. It is advisable to employ cognitive assessments as a tool for esports selection, taking into account not only ranking but also general cognitive abilities. (b) Enhancing the importance of attention and spatial assessments. Simply conducting systematic cognitive tests is not sufficient. Tests such as selective reaction time and visual working memory tasks prove highly valuable (*Williams & Davids, 1995*; *Tanaka et al., 2013*; *Pluss et al., 2020*). (c) Incorporating anticipation and decision-making tasks in esports selection or training. Anticipation and decision-making are highly significant in mainstream sports research

(*Williams et al., 2006*; *McGee & Ho, 2021*). Future studies and practices should include tasks that assess anticipation and decision-making abilities in esports.

### Funding

This work was funded by the Open Funding Project of National Key Laboratory of Human Factors Engineering (614222202062211), Lizhong Chi is the Grant Recipient; the 2023 Top Graduate Innovative Research Program of Beijing Sport University (2023018) supported the APC for this article. The funders had no role in study design, data collection and analysis, decision to publish, or preparation of the manuscript.

### Grant Disclosures

The following grant information was disclosed by the authors:
The Open Funding Project of National Key Laboratory of Human Factors Engineering: 614222202062211.

### Competing Interests

The authors declare there are no competing interests.

### Author Contributions

- Haofei Miao conceived and designed the experiments, performed the experiments, prepared figures and/or tables, authored or reviewed drafts of the article, and approved the final draft.
- Hao He conceived and designed the experiments, performed the experiments, analyzed the data, prepared figures and/or tables, authored or reviewed drafts of the article, and approved the final draft.
- Xianyun Hou conceived and designed the experiments, authored or reviewed drafts of the article, and approved the final draft.
- Jinghui Wang performed the experiments, authored or reviewed drafts of the article, and approved the final draft.
- Lizhong Chi conceived and designed the experiments, authored or reviewed drafts of the article, and approved the final draft.

### Human Ethics

The following information was supplied relating to ethical approvals (i.e., approving body and any reference numbers):

The present meta-analysis does not require an ethical statement.

### Data Availability

The raw data, code and other supplementary materials are available at OSF: Miao, Haofei, Hao He, and Lizhong Chi. 2023. "Cognitive Expertise in Action Video Games Experts: A Meta-analysis." OSF. February 24. The following link leads directly to the public data and code: https://osf.io/kdfn2/.

## Supplemental Information

Supplemental information for this article can be found online at http://dx.doi.org/10.7717/peerj.17857#supplemental-information.

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
