# Peer review of "Cognitive expertise in esport experts: a three-level model meta-analysis"

_PeerJ, doi:10.7717/peerj.17857_

## Round 0.1 · original submission · Major Revisions

Dear Author:

Thank you for submitting your manuscript to PeerJ Journals. After several reviews, we consider that some improvements are needed. Please read and respond to the reviewers.

Regards

Dr. Manuel Jiménez

Reviewer 1 ·

Basic reporting

The article tries to determine the cognitive characteristics that differentiate esports experts from amateurs.

The paper is well written and the cited literature is up to date.
The text provides a good introduction and justifies the convenience of the study. The results are well described and the tables, figures and supplementary material are adequate.The discussion and conclusions are adjusted to the results obtained.

Experimental design

The study conforms to the journal's Aims & Scope, highlighting its novelty and interest for the field of video games, halfway between cognitive psychology, sports psychology and human-computer interaction.
The objectives of the study and the proposed hypotheses are relevant. The method used is appropriate, well executed and the necessary information is provided to be replicated.

Validity of the findings

The results are relevant to the extent that they allow us to review the most commonly used measures to evaluate cognitive expertise in esports. Also noteworthy is the evidence they provide about the cognitive dimensions that are truly discriminatory when selecting and training talented gamers. In this sense, the conclusions offered by the authors are clear and precise, helping future researchers to use them, both for their application and for new studies.
The precision in the use of inclusion criteria for the study is notable, especially in the definition of experts versus amateurs, as well as the decision not to include casual gamers.

Additional comments

However, it would be convenient that the authors justify the statements about “extraordinaty emergence in growth over the past two decades” (line 45), and its development through Olympic Virtual Series (lines 46-47). Another statement that should be justified is the one related to the “level of proficiency within just or two years of game experience” (line 57).

·

Basic reporting

The authors of the study titled "Cognitive expertise in esport experts: A three-level model meta-analysis" present an interesting meta-analysis of key cognitive skills with a good justification of the need for the study. The references are adequate, up-to-date and allow us to adequately contextualize the current situation in esports and its ecosystem. The figures and tables are adequate, provide information and the results are well described for the objectives set in the meta-analysis. Finally, the discussion is well presented and offers interesting future lines of study.

Experimental design

The study adequately adjusts to the aims and scope of the journal. The hypotheses and objectives of the study presented are interesting, in addition to providing substantial value to the field of study of video games and esports, evaluating the relevant interest that exists between levels of expertise and cognitive skills, allowing the development of future recruitment programs. and talent development, an underdeveloped field in esports. In the method they explain in detail how they have carried out the different steps and it can be replicated.

Validity of the findings

The results presented in the present manuscript are relevant and accurate. They clearly explain the calculations they have used, guaranteeing a correct approach and statistical analysis. It is necessary to highlight the findings found in which variables best discriminate the degree of expertise, thus guiding future research in the field that allows contrasting the results obtained. The conclusions offered are well justified by the results presented.

Additional comments

In introduction, in line 48-49 I suggest more justification of why esports are an opportunity to explore and research expertise. In line 56-58 when talking "Moreover, some esports players are able to attain an
57 internationally recognized level of proficiency within just one or two years of game experience, in stark
58 contrast to the prolonged practice periods typically required in other domains of expertise" I suggest to explain better this statement, because esports players usually they play from the 12/13 years old and with 17/18 they starts compiting and they need this 2 years to arrive to international tournaments, so I will recheck that conclusion or at least justifiy it.
.

·

Basic reporting

Introduction
Line 42-44: This sentence doesn’t read well, consider breaking into multiple sentences.

Line 46: Delete period after (IOC)

Line 56-58: While I appreciate the point the authors are trying to make, I don’t believe this rapid rate of expertise development is typical in most gamers and should there for not be discussed as if it is the norm. However, if the authors believe this is the case, please provide references to justify the statement or provide examples. Additionally, when discussing expertise and practice regarding gaming, I don’t believe that years of experience is an appropriate measure of time as gamers often play and practice significantly more hours per day than traditional sports or skills so discussing experience in years does not provide an accurate representation of time spent learning.

Line 85: worthing should be worth noting?

Line 116: selection should be selecting?

Methods
Line 143: How may investigators were involved in the search and screening process? Please describe the process or article assessment and how were disagreements resolved?

Discussion
Line 348 – 351: The classifications of cognitive ability should be discussed or listed earlier in the manuscript as these are discussed but not mentioned earlier in the manuscript. These cognitive abilities provide important context to the study design and analyses performed.

Experimental design

The research question proposed by the authors was well defined and very appropriate. The justification given for their approach is well founded and takes an excellent approach to answering an important question. As gaming is multifaceted and highly variable, utilizing one metric to define cognitive performance would be ineffective, therefore the authors’ approach is much more insightful. Additionally, with the growing popularity of esports, understanding the cognitive differences between experts and amateurs will aid those in leadership roles and continue to improve the industry. However, there are a number of areas that need significant improvement or justification.

Early in the introduction, the authors discuss the various cognitive demands of gaming, stating “a wide array of cognitive demands such as perception, attention, memory, planning, and decision-making” However, a significant emphasis is later placed on the assessment of speed and reaction time. These metrics, while partially cognitive in nature, also involve a significant motor aspect and are often used to assess motor skills. Please provide justification that these metrics are relevant outcome measures for cognitive ability. If there metrics are unique to cognitive based tests, please clarify what is meant by speed and reaction time with respect to those dependent variables.

Line 161: Please add the search terms used to perform your literature search

Line 197: Please provide justification for choosing these three game genres at moderator variables.

The methods section requires greater clarity and is overcomplicated, in particular, the data coding and data analysis sections. The information provided by the authors does not follow a natural progression and makes it difficult to understand how analysis was performed. Please provide a straightforward explanation of the analysis methods used and why each analysis method was used. In particular, the moderator analysis causes some confusion. Please explain exactly what information the moderator analysis is providing and what the relationship is between the moderator variables and the dependent and independent variables. As I understand it, this study is exploring the relationship between level of esports expertise (professional level) and cognitive abilities which is represented by an over all effect size. Therefore, how is professional level used as a moderator variable? This confusion may be due to a lack of experience on my part, but I believe that the average reader may also find this analysis confusing.

Validity of the findings

Findings from this study appear valid and meaningful. I believe these data will provide highly impactful information to those in the esports industry and provide valuable information to those developing in esports. I believe the only issues with this study exist in the writing of the methodology and justification of the analyses chosen to investigate the data.

---

## Round 0.2 · Major Revisions

Dear Authors:

Two of the reviewers have accepted the current format. Still, the third reviewer has presented a series of issues that, in my opinion, are relevant for the manuscript to reach quality standards that will make it very important within the field of study. I understand that you are awaiting the publication of the manuscript as soon as possible. Still, I am sure that you will understand the importance of achieving a study that is as spherical as possible.

Please pay attention to the last reviewer's comments and answer his questions.

Best regards
Dr. Manuel Jimenez

Reviewer 1 ·

Basic reporting

Now the authors have replied to the reviewers and the article has already been substantially improved, being clearer and more precise.

Experimental design

No comment.

Validity of the findings

No comment.

Additional comments

A nice contribution to the literature.

·

Basic reporting

All suggestion has been aswered.

Experimental design

All suggestion has been aswered.

Validity of the findings

All suggestion has been aswered.

Additional comments

All suggestion has been aswered.

·

Basic reporting

The authors present a meta-analysis of cognitive skills in pro vs amateur gamers. This is a very interesting study which asks a different and complementary question to other existing meta-analyses of video game players vs non-players or those assessing the cognitive effects of video game training. The Inclusion of gray literature is a real strength especially given the small number of studies in the field and their heterogeneity in terms of criteria for defining pro players (and amateurs) and their variability in cognitive tasks and designs. While this study would make a novel contribution to the literature, it suffers from several weaknesses that need to be addressed to strengthen the manuscript and make it acceptable for publication in PeerJ. These points are detailed below.

Overall, the main issue is that the method section lacks a lot of critical details to understand and evaluate the quality of the methods, in particular regarding the coding and analysis. It would greatly help if the authors would agree to share their data and code, in accordance with PeerJ policy.

There are several additional important points that require clarification.

First, the preregistration is inaccessible. The OSF link line 147 requires approval which makes the review not anonymous. When access has been granted, the content of the preregistration is insufficient to consider it a pre-registered study. The hypothesis and analysis strategy are not transparently described. The most important information was updated only recently and probably after a first manuscript had been submitted. Critical information about search strategy, inclusion/exclusion criteria, coding and analysis strategy are not detailed enough. As a reviewer,I found it impossible to “[...] understand the registered intention of the study and consider important deviations, omissions, and analyses that were not preregistered in assessing the work.”
https://statmodeling.stat.columbia.edu/2023/11/16/open-letter-on-the-need-for-preregistration-transparency-in-peer-review/

Second, the authors need to clarify their definition and coding of game genres. The readers would benefit from a description of the different game genres as there is no consensus on the definition and the categorisation of games into distinct genres. Moreover, there seems to be some confusion in their categories. For example, they seem to confuse RTS and MOBA. On p16/37 rows 434-435 they write “These findings are consistent with previous research indicating no distinctions in attention and multi-target tracking abilities between FPS and RTS amateurs (Bickmann et al., 2021).” The study by Bickmann et al talks about MOBA, not RTS. Also in their methods they say “The most common esports genres were Real-Time Strategy (RTS) game (studies k = 6) and FPS game (studies k = 2), while some studies combined several game genres or included other game genres (studies k = 7).” They categorized according to Pedraza-Ramirez et al. (2020) but they didn’t respect the categorization. More generally, upon checking it appears that there were
2 RTS studies: Kang et al. 2020 (Starcraft); Kim et al. 2023 (Starcraft - SC),
10 MOBA studies (studies involving League of Legends, DOTA, Heroes of the Storm (HotS)
6 FPS studies involving CS, CSGO, Overwatch, PUBG (excluding PUBG Mobile),Battlefield, Rainbow6, COD
Other genres:
2 Fighting games studies: Phillips et Green 2023; Tanaka et al 2013
1 Rhythm game study: Phillips & Green 2023
1 PUBG mobile study: Wang & Zhang 2019
1 Card game: Artifact Wang & Zhang 2019
1 Racing game: Popkart Wang & Zhang 2019

Experimental design

Third, the method section lacks critical details required to evaluate the meta-analytic procedures. In particular
1. There are 147 effects from 15 studies. How was the dependence structure modeled? Were the effect sizes from the same study considered hierarchically related or correlated or both? I don’t understand the 3-levels. I see only 2 levels: effect sizes and studies. Can you please clarify? My understanding is that level zero is not actually modeled and refers to measurement errors.

2. How did you verify selection criteria?In particular expertise criteria: (i) registered pro players or (ii) part of an organized team or (iii) top 1% amateur ?
How was expertise CODED ?
Same for amateurs… how did you ensure they never played in a tournament? They could have low ranking now but could be in the top 1% in the past?
3. How did you ensure that non-gamers were excluded?
4. How did you assess/classify the game genre? Please, provide the K and some examples of games in each category. Where are MMORPGs (e.g., WOW)?
5. Note that Hearthstone is not a game genre but a game title.
6. I looked for “Popkart” and only found the game “Crazyracing Kartrider” related to the search.
7. The authors consider Battleground and PUBG as 2 different games (because in one study they called it Battleground but it should also mean PUBG, just don’t know whether it is PUBG mobile or not) which is confusing. This information must be present in the primary studies although with different definitions and criteria. It would be important to know how much consistency there was in the reporting of this

8. How many data points were estimated using GetData Graph Digitizer?

9. Moderator analysis: Were moderators tested individually or were all moderators included in the model?
9a. What is a “professional level as a moderator”? The effect is not significant (line 303) but the comparisons show important differences in cognition between professionals and amateurs (please provide, k and m as well as CI), whereas no difference is seen between high-ranked and amateurs (add k, m and CI). How were the “amateurs” defined in these 2 studies? Could someone be both professional and high-ranked or only fit one criterion, not the other (i.e., only professional or only high-ranked)?

9b. I was surprised to see a non-significant effect for perception favoring the amateurs (negative direction). I would greatly appreciate it if the authors could share the data and code. Or at least provide a table with all the effect sizes and tasks, and dependent measures as well as the means and SDs per group (+ references) included in “perception” – and in the other categories too?

9c. Similarly, I was surprised that the effect on SPEED was not significant. I would like to see a table describing how tasks and measures were coded as speed and included in this analysis, but also, and more importantly (i) which cognitive domain and (ii) which expertise group they come from.
The analysis of game genres is particularly puzzling. It is striking that none of the game genres showed significant effect, when the overall effect was significant and most likely driven by the heterogeneous OTHER category.

9d. Why consider gender at the study level? Were gender matched between experts and amateurs in all studies? Please consider coding whether groups were matched for gender or which group had more males?
Age is significant. What is the range? Does age correlate with expertise level? Could this effect come from accumulated experience in experts or from the amateur group performing worse with age? (see discussion lines 425-430). It would be interesting if the authors could plot the normalized performance data for each group to see if this effect comes from the pro gamers performing better with age or from the amateurs performing poorer with age.

10. Sensitivity analysis. Did the authors apply leave-one-out by excluding a study or an effect size in each iteration? Could the authors show the distribution of effects obtained?

11. Risk of bias. How was the overall study quality on average and how variable was it between studies ? The authors only mention “acceptable” on line 354 but more details (a table or figure) would be appreciated. Were there differences between studies based on how expertise was defined (e.g., pro vs high rank)?

12. Lines 405: “selective RT falls into bottom-up attention”. What is selective reaction time? Please, give examples? Why is it coded in bottom-up?

13. Implication c (incorporating anticipation and decision making) does not stem from the study results and should thus be rephrased to indicate that it was missing from the reviewed studies.

Validity of the findings

It is difficult to assess the validity of the findings because (1) important methodological details are missing, (2) data and analysis code are not shared and (3) the preregistration does not provide enough details.

Additional comments

One reference is missing and seems to correspond to a conference abstract which has no stats reported. How did the authors obtain the data https://escholarship.org/uc/item/3ng4v5b2#main. Was this considered as published (peer-reviewed) or unpublished? By contrast, this review from the same author which contains other references is not cited – and may suggest the authors may have other unpublished work relevant to this meta-analysis https://escholarship.org/uc/item/9nt8j13s.
The criteria for defining pro players vs top 1% players and more importantly the comparison between these groups is a hard one. I am not sure if the same criteria apply and are comparable across all games. For example, CSGO has a world rank, whereas LoL has many ranks (one for each server). And authors of a primary study indicate “top 1%”, then diamond 1 wouldn’t qualify for example in West Europe because they are a bit higher than top 1% (if they want to fully respect a criteria).

Minor issues
They misspelled the game Artifact (they wrote Aritifact).

---

## Round 0.3 · accepted · Accept

Dear Authors:

Thank you for being so patient. The manuscript has been improved and accepted for publication on PeerJ.

Best Ragards

Dr. Manuel Jiménez

Reviewer 1 ·

Basic reporting

No comment

Experimental design

No comment

Validity of the findings

No comment

Additional comments

Now the authors have replied to the reviewers and the article has already been substantially improved, being clearer and more precise.

·

Basic reporting

After reading again the full manuscript an evaluate the changes during the different issues raised by the reviewers all suggestions have been answered.

Experimental design

After reading again the full manuscript an evaluate the changes during the different issues raised by the reviewers all suggestions have been answered.

Validity of the findings

After reading again the full manuscript an evaluate the changes during the different issues raised by the reviewers all suggestions have been answered.

Additional comments

After reading again the full manuscript an evaluate the changes during the different issues raised by the reviewers all suggestions have been answered.